# Dually Responsive Poly(*N*-vinylcaprolactam)-*b*-poly(dimethylsiloxane)-*b*-poly(*N*-vinylcaprolactam) Polymersomes for Controlled Delivery

**DOI:** 10.3390/molecules27113485

**Published:** 2022-05-28

**Authors:** Veronika Kozlovskaya, Yiming Yang, Fei Liu, Kevin Ingle, Aftab Ahmad, Ganesh V. Halade, Eugenia Kharlampieva

**Affiliations:** 1Department of Chemistry, The University of Alabama at Birmingham, Birmingham, AL 35294, USA; vkozlovs@uab.edu (V.K.); yy930924@outlook.com (Y.Y.); yiyaoleo@gmail.com (F.L.); 2Department of Cardiovascular Diseases, School of Medicine, The University of Alabama at Birmingham, Birmingham, AL 35294, USA; kaingle@uab.edu; 3Department of Anesthesiology and Perioperative Medicine, School of Medicine, The University of Alabama at Birmingham, Birmingham, AL 35294, USA; aftabahmad@uabmc.edu; 4Division of Cardiovascular Sciences, Department of Internal Medicine, University of South Florida, Tampa, FL 33602, USA; ghalade@usf.edu; 5The Center for Nanoscale Materials and Biointegration, Department of Physics, The University of Alabama at Birmingham, Birmingham, AL 35294, USA; 6The O’Neal Comprehensive Cancer Center, School of Medicine, The University of Alabama at Birmingham, Birmingham, AL 35294, USA

**Keywords:** polymersome, degradable, poly(*N*-vinylcaprolactam), temperature-responsive, in-vivo toxicity

## Abstract

Limited tissue selectivity and targeting of anticancer therapeutics in systemic administration can produce harmful side effects in the body. Various polymer nano-vehicles have been developed to encapsulate therapeutics and prevent premature drug release. Dually responsive polymeric vesicles (polymersomes) assembled from temperature-/pH-sensitive block copolymers are particularly interesting for the delivery of encapsulated therapeutics to targeted tumors and inflamed tissues. We have previously demonstrated that temperature-responsive poly(*N*-vinylcaprolactam) (PVCL)-*b*-poly(dimethylsiloxane) (PDMS)-*b*-PVCL polymersomes exhibit high loading efficiency of anticancer therapeutics in physiological conditions. However, the in-vivo toxicity of these polymersomes as biocompatible materials has not yet been explored. Nevertheless, developing an advanced therapeutic nanocarrier must provide the knowledge of possible risks from the material’s toxicity to support its future clinical research in humans. Herein, we studied pH-induced degradation of PVCL_10_-*b*-PDMS_65_-*b*-PVCL_10_ vesicles in-situ and their dually (pH- and temperature-) responsive release of the anticancer drug, doxorubicin, using NMR, DLS, TEM, and absorbance spectroscopy. The toxic potential of the polymersomes was evaluated in-vivo by intravenous injection (40 mg kg^−1^ single dose) of PVCL_10_-PDMS_65_-PVCL_10_ vesicles to mice. The sub-acute toxicity study (14 days) included gravimetric, histological, and hematological analyses and provided evidence for good biocompatibility and non-toxicity of the biomaterial. These results show the potential of these vesicles to be used in clinical research.

## 1. Introduction

Limited tissue selectivity and the lack of targeting of anticancer therapeutics during systemic administration can result in deleterious side effects of anticancer chemotherapy [1,2,3,4]. To improve the efficacy of anticancer drugs, polymer-drug delivery carriers, including polymer conjugates, nanoparticles, and micelles, have been used to shield the therapeutics and prevent their premature release in healthy tissues [2,3,4,5,6,7]. Among those, polymer vesicles assembled from block copolymers, i.e., polymersomes, have been recognized as effective nanocarriers because of their cell-mimetic membranes, improved colloidal and mechanical stability, and increased efficiency in drug entrapment, unlike liposomes [5,6,7,8,9,10,11,12,13,14].

The stimuli-sensitive block copolymers have been used to imprint a selective response from a polymer vesicle with a programmable stimuli-triggered behavior [15,16]. Stimuli-sensitive polymersomes can release encapsulated therapeutics upon stimuli-triggered changes in the vesicle membrane, resulting in a partial or complete disassembly of a vesicle, leading to a burst or sustained release of a drug [17,18,19,20,21]. The use of dually-responsive polymersomes that are assembled from temperature- and pH-sensitive block copolymers is particularly interesting due to the slightly acidic pH (pH 6.8–5) [22] and elevated temperature (T = 38–42 °C) [23] conditions encountered in the tumor, inflamed tissues, and in the endosomal and lysosomal intracellular compartments [24].

The changes in a nanovesicle membrane upon pH lowering have been shown as an efficient stimulus to selectively release anticancer drugs in tumor cells and tissues [25,26,27]. For example, Armes and coworkers reported pH-sensitive polymersomes based on poly(2-(methacryloyloxy) ethyl phosphorylcholine)-*b*-poly(2-(diisopropylamino) ethyl methacrylate) for controlled release of doxorubicin (DOX) and DNA delivery at pH = 6 [26,27]. Borchert et al. reported a quick dissolution of poly(2-vinylpyridine)-*b*-poly(ethylene glycol) (PEG) polymersomes at the pH drop from 7.4 to 5.5, triggering a fast release of 4-methoxy coumarin fluorescent dye [28]. Although the close-to-physiological pK_a_ of the membrane copolymer was exploited for the disassembly of a non-degradable membrane, [10] degradable polymersomes that produce nontoxic byproducts of low molecular weight (<20 kDa) would be more desirable for systemic delivery.

To address that, the degradation of polymersomes due to the pH-induced hydrolysis of polyester or polycarbonate blocks has been explored [29,30,31]. Feijen and coworkers investigated polymer vesicles from the block copolymers of PEG and biodegradable polyesters or polycarbonates as artificial cells that released fluoresceinamine upon degradation of ester bonds [29]. Lee and coworkers reported biodegradable polymersomes of poly(2-hydroxyethyl aspartamide) grafted with lactic acid oligomers [32]. The biodegradable polymersomes of PEG-*b*-poly(lactic acid) and PEG-*b*-poly(caprolactone) have been shown to release DOX as a result of acidic hydrolysis [30]. These anticancer drug-loaded polymersomes with the degradable polyester and polycarbonate blocks have also been shown to be capable of decreasing tumor volumes [29,32]. The use of the hydrazone linker between the blocks in poly(butadiene)-*b*-PEG diblock copolymer afforded a shedding of the hydrophilic corona at a slightly acidic pH of 6.4 without disrupting the colloidal stability of the vesicles [33,34]. Enzymatically degradable polymersomes from chitosan-*g*-[poly(l-lysine)-*b*-poly(ε-caprolactone)] copolymer have been shown by Bourgat et al., where both enzymatic degradation of poly(l-lysine) and degradation of poly(caprolactone) due to the hydrolysis at acidic pH were demonstrated [35].

We have demonstrated a temperature-sensitive polymersome assembled from poly(*N*-vinylcaprolactam)_n_-*b*-poly(dimethylsiloxane)_m_-*b*-poly(*N*-vinylcaprolactam)_n_-(PVCL_n_-PDMS_m_-PVCL_n_) triblock copolymer [36]. PDMS and PVCL are both biocompatible and generally not cytotoxic [37,38,39]. PVCL is a temperature-sensitive polymer with a phase transition in the range of 36 to 50 °C. The development of the PVCL-based self-assemblies for controlled drug delivery has been gaining increasing attention in recent years [40,41,42]. We showed that the PVCL-PDMS-PVCL polymersomes reversibly decreased in size without changing their shape upon the solution temperature increase in the range of 37 to 42 °C due to the lower critical solution temperature (LCST) transition of the PVCL blocks [36]. The observed temperature-induced size decrease of the vesicle was also accompanied by a decreased thickness of the vesicle membrane, as revealed by small-angle neutron scattering (SANS) [43]. These temperature-induced structural changes of the PVCL-PDMS-PVCL membrane have been shown useful for a precisely controlled sustained release of encapsulated DOX from these vesicles in the physiologically relevant temperature range of 37–42 °C. The DOX release was found to be concentration- and time-dependent and led to more than 95% cytotoxicity to human alveolar adenocarcinoma cells [36]. 

We also showed an extraordinarily high loading capacity of DOX (49%) and encapsulation efficiency (95%) from aqueous DOX by the PVCL-based polymersomes assembled from poly(3-methyl-*N*-vinylcaprolactam)_58_-*b*-PVPON_n_ (PMVCL-PVPON) diblock copolymer due to hydrophobic interactions between drug molecules and PVCL-based blocks at room temperature [44]. In our in-vivo study, the comparison of these PVCL-based polymersomes with pegylated liposomes for encapsulation of DOX, we found that injection of liposome-encapsulated DOX caused organ toxicity in contrast to the polymersome-encapsulated DOX and saline control [44]. Despite that, the PMVCL-PVPON polymersomes were not biodegradable, unlike the PVCL-PDMS-PVCL triblock copolymer vesicles, which could be disassembled at low pH due to the hydrolysis of the ester bond between PVCL and PDMS blocks [36].

Conversely, despite the high loading efficiency of toxic DOX and both pH- and temperature-response in the physiological conditions, the in-vivo toxicity of the degradable PVCL-PDMS-PVCL polymersomes as biocompatible material has not yet been explored. However, the development of an advanced therapeutic nanocarrier must provide information on some possible risks from material toxicity to support its clinical research in humans [45]. Therefore, in this work, we studied the pH-responsive degradation of the PVCL-PDMS-PVCL vesicles and the pH-triggered release of DOX encapsulated by these polymersomes. We used polymer vesicles made from poly(2-methyl-2-oxazoline)_14_-poly(dimethylsiloxane)_65_-poly(2-methyl-2-oxazoline)_14_ (PMOXA_14_-PDMS_65_-PMOXA_14_) amphiphilic triblock copolymer as a non-pH-responsive and nondegradable control. Meier and coworkers demonstrated PMOXA-PDMS-PMOXA-based polymersomes as nanocarriers for encapsulation of drugs, enzymes, proteins, and nucleic acids [46,47,48]. This triblock copolymer forms highly mechanically stable carriers ranging in size from 50 to 300 nm with impermeability to small molecules in aqueous media. Passive channel proteins have been embedded within the PMOXA-PDMS-PMOXA vesicle wall to allow for the permeability of the vesicle membrane [13]. Yet, the channel protein’s processing and embedding in the vesicle membrane can be complicated and costly.

As the PVCL-PDMS-PVCL polymersomes can respond to temperature variations and can be degraded in the body, we investigated the in-vivo toxicity of the PVCL_10_-PDMS_65_-PVCL_10_ polymersomes using a mouse model (Figure 1). The in-vivo toxicity of the polymer vesicles was evaluated through the sub-acute study for 14 days by the necropsy and histology analyses of major animal organs, including spleen, heart, and kidney analysis, and hematology profile measurements. Our results indicate that PVCL_10_-PDMS_65_-PVCL_10_ polymersomes with the dually induced release of DOX are degradable under physiologically relevant conditions and have good biocompatibility, which demonstrates strong potential for further development of these vesicles as therapeutic nanocarriers for future anticancer clinical treatment.

## 2. Results and Discussion

### 2.1. Synthesis of PVCL_10_-PDMS_65_-PVCL_10_ and PMOXA_14_-PDMS_65_-PMOXA_14_ Triblock Copolymers and Vesicle Assembly

The triblock copolymer PVCL_10_-PDMS_65_-PVCL_10_ was synthesized as we described earlier [36,43]. After synthesis and purification, the number-average molecular weights of bis(hydroxyethyl)oxypropyl poly(dimethylsiloxane) (PDMS_65_) were obtained from ^1^H-NMR analysis and calculated from the ratio between the integrals at δ = 0.3 ppm (-SiCH_3_ protons in the PDMS block repeat units) and at δ = 0.6 ppm (the methylene protons in the end groups). The length of the PVCL block was determined by comparing integrals at δ = 4.5 ppm (>C*H*- on the PVCL block end groups) and at δ = 0.3 ppm (the methyl protons for -Si-C*H*_3_ in PDMS). The length of each PVCL was half of all PVCL repeat units (Figure 2a). As a non-responsive counterpart, a triblock copolymer of PMOXA_14_-PDMS_65_-PMOXA_14_ was synthesized via cationic ring-opening polymerization of 2-methyl-2-oxazoline using PDMS_65_-triflate as described previously by Meier and coworkers [46]. The length of the PMOXA block was calculated from the ^1^H-NMR spectra of the copolymer by comparing the ratio between the integrals at δ = 0 to δ = 0.3 ppm (SiC*H*_3_ in PDMS) and at δ = 0.6 ppm (the methylene protons in the end groups) to integrals at δ = 2.1 ppm (the methyl group of MOXA) [43].

The copolymer vesicles were assembled from PVCL_10_-PDMS_65_-PVCL_10_ and PMOXA_14_-PDMS_65_-PMOXA_14_ triblock copolymers through the nanoprecipitation method as we demonstrated earlier [36,43,49]. The anticancer drug DOX was encapsulated by the vesicles during the block copolymer assembly. For that, the copolymer ethanol solution (5 mg mL^−1^) was added to a vigorously stirred 1 mg mL^−1^ DOX buffer solution (0.01 M phosphate). After 6-h assembly and DOX encapsulation, the vesicles were purified by dialysis in a 0.01 M buffer solution at pH = 7.4 to remove ethanol and free, nonencapsulated DOX.

Transmission electron microscopy (TEM) imaging of the DOX-loaded PVCL_10_-PDMS_65_-PVCL_10_ (Figure 2b) and PMOXA_14_-PDMS_65_-PMOXA_14_ (Figure 2c) polymersomes dried on TEM grids confirmed the formation of spherical vesicular morphology for both block copolymers. As the polymersomes were negatively stained with uranyl acetate dye for TEM contrast, they demonstrated a hollow non-stained vesicle interior due to the dye impermeability through the vesicle membrane. The DOX-free PVCL_10_-PDMS_65_-PVCL_10_ vesicles had an average hydrodynamic diameter of 396 ± 65 nm in the buffer solution at pH = 7.4. The hydrodynamic size of the DOX-loaded PVCL_10_-PDMS_65_-PVCL_10_ polymersomes in the buffer solution was slightly larger and was 470 ± 72 nm as measured by DLS. These results agree with our previous data where PVCL_10_-PDMS_65_-PVCL_10_ produced vesicles with an average hydrodynamic size of 530 nm (with polydispersity index, PDI = 0.054) after 24 h of nanoprecipitation self-assembly and were stable for at least 72 h as measured by DLS in water [36]. A slight size increase of the PVCL-PDMS-PVCL vesicles after DOX encapsulation compared to DOX-free polymersomes can be due to the reduced solubility of DOX hydrochloride in the initial ethanol/water solution [50]. After the copolymer assembly via the hydrophobic association, a possible hydrophobic association of DOX hydrochloride with PVCL could lead to the shrinkage of a vesicle. However, the removal of the ethanol via dialysis in the buffer solution can facilitate the dissociation of DOX from the outer PVCL chains, followed by the vesicle swelling and, consequently, a slight increase in the vesicle diameter. 

The control nonresponsive block copolymer PMOXA_14_-PDMS_65_-PMOXA_14_ also produced the vesicular morphology, as evidenced by TEM analysis in Figure 2c. The average hydrodynamic size of the DOX-loaded PMOXA_14_-PDMS_65_-PMOXA_14_ polymersomes was 127 ± 70 nm as measured by DLS in the 0.01 M phosphate buffer solution at pH = 7.4. Relatively broad size distribution for PMOXA_14_-PDMS_65_-PMOXA_14_ vesicles is mainly due to the nanoprecipitation assembly method, unlike the thin film hydration method where the polymersomes exhibit a narrower size distribution because of the sample extrusion through a nano-porous membrane as we demonstrated earlier [43]. The PMOXA_14_-PDMS_65_-PMOXA_14_ vesicles were stable in solution at pH = 7.4 for at least 3 h, as confirmed by DLS analysis (Figure 3, squares, and circles). However, within the 12-h period, we observed a slight size decrease to an average diameter of 91 ± 60 nm, which was accompanied by increased scattering intensity at a micrometer size range (>1 µm) (Figure 3, triangles). This result can be explained by a slow aggregation of the vesicles with time at this pH and settling down, as reported recently [51], which agrees with the data on the average size and size distribution decrease.

### 2.2. pH-Induced Degradation of PVCL_10_-PDMS_65_-PVCL_10_ Polymersomes

The stability of both polymersomes under acidic conditions was also assessed. For that, the vesicles were assembled in the 0.01 M phosphate buffer solution at pH = 3, followed by dialysis in the buffer at 37 °C for 12 h with the vesicle size analyzed by DLS after 0, 3, and 12 h of incubation. The DLS analysis of the PMOXA_14_-PDMS_65_-PMOXA_14_ vesicles at pH = 3 (Figure 4a) demonstrates that the vesicles showed good stability under this condition for at least 12 h. The hydrodynamic size of these polymersomes was constant at pH = 3, with an average diameter of 123 ± 73 nm within an hour of preparation, 106 ± 68 nm after 3-h incubation, and 106 ± 60 nm after 12-h incubation. This result agrees well with the PMOXA-PDMS-PMOXA chemical composition as it is neither a pH-responsive nor pH-degradable polymer where the PMOXA blocks are linked to the PDMS block through the non-degradable ether bond.

In contrast, the PVCL_10_-PDMS_65_-PVCL_10_ polymersomes assembled and then incubated at pH = 3 showed a time-dependent size decrease as analyzed by DLS (Figure 4b). The average hydrodynamic diameters of PVCL_10_-PDMS_65_-PVCL_10_ vesicles decreased from 396 ± 88 nm within an hour after assembly, to 295 ± 30 nm after 3-h incubation, to 142 ± 20 nm after 5-h incubation, and to 63 ± 23 nm after 12-h incubation at pH = 3 and 37 °C. Apparently, the incubation and dialysis of the vesicles in acidic solutions resulted in hydrolysis of the ester linkages between the PVCL and PDMS blocks and the copolymer vesicle dissolution. We observed an almost complete disappearance of the vesicle solution turbidity after 12 h of the incubation, which implied a total disassembly of the polymersomes.

The dissolution of the PVCL_10_-PDMS_65_-PVCL_10_ vesicles was much faster than the degradation of PEG-PCL or PEG-PLA diblock copolymer polymersomes, reported earlier to achieve 80% degradation within 90 h of incubation under acidic conditions [30]. During the hydrolysis, the polyester-based PCL and PLA blocks lose their hydrophobicity gradually due to pH-catalyzed hydrolysis, and the vesicles break down gradually through the formation of a porous membrane [30]. Moreover, as the polyester-based hydrophobic block forms a middle layer of the vesicle membrane which repels water because of its hydrophobicity, the degradation kinetics of the PCL- and PLA-based vesicles are usually very slow [29]. For PVCL_10_-PDMS_65_-PVCL_10_ vesicles, the ester bond between the hydrophobic PDMS and hydrophilic PVCL blocks is easily accessible due to PVCL hydrophilicity at 25–37 °C [52], and the hydrolysis of the ester link can be achieved faster, which facilitates a relatively quick disassembly of the vesicles within 12 h.

In addition, we previously showed, using small-angle neutron scattering [43], that the coil-to-globule phase change of the PVCL block above 37 °C could trigger rearrangement of the PVCL chains that could interact with PDMS blocks due to their LCST-induced hydrophobicity, creating a pathway for small molecules to pass through the PDMS hydrophobic membrane at elevated temperatures (T > 37 °C). Thus, for example, about 20% of encapsulated DOX could be released from PVCL_10_-PDMS_65_-PVCL_10_ vesicles incubated at 37 °C for 10 h [36]. This behavior could also be favorable for accelerated hydrolysis of the ester linkage between PDMS and PVCL blocks. In addition, small hydrolysis products can be removed from the vesicle interior due to the leaking membrane at higher temperatures (T > 37 °C).

### 2.3. pH-Dependent Release of DOX from PVCL_10_-PDMS_65_-PVCL_10_ Vesicles

In our previous work, we demonstrated that by increasing the solution temperature from 30 to 45 °C, DOX could be gradually released from the DOX-loaded PVCL_10_-PDMS_65_-PVCL_10_ vesicles [36]. While negligible at T < 30 °C, the cumulative DOX release increased from 26% to 76% at T = 35 °C and T = 40 °C, respectively, and reached 87% at 45 °C [36]. In this work, we studied DOX release from PVCL_10_-PDMS_65_-PVCL_10_ vesicles at varied pH conditions. 

After the assembly of the vesicles using 0.5 mg mL^−1^ DOX solution, the free, non-encapsulated drug was dialyzed off along with ethanol during the dialysis at pH = 7.4. The vesicles showed an excellent drug loading efficiency of 45% for DOX hydrochloride with a loading capacity of 40%, which is similar to our previous results [36]. The release studies were carried out at 37 °C and involved varying 0.01 M phosphate buffer solutions with varied pH values. At pH = 7.4, the encapsulated DOX was fully sustained in the vesicles for at least 7 days with less than 9 ± 2% DOX release, which can be attributed to the freeing of loosely outer shell-bound DOX (Figure 5a, diamonds). The study revealed a significant amount of DOX released from the PVCL_10_-PDMS_65_-PVCL_10_ vesicles upon the pH decrease from the physiological pH (Figure 5a). Thus, after 24-h incubation, the amount of released DOX from the polymersomes was found to be 9 ± 2%, 22 ± 2%, and 96 ± 4% DOX at pH = 7.4, pH = 4, and pH = 3, respectively. These drug release data agree well with the vesicle degradation results (Figure 4b), where the vesicle size drastically decreased to 63 ± 23 nm after 12 h of incubation at pH = 3. These DOX release results also confirm that the vesicle degradation mainly occurs in the pH range from 4 to 3, which is characteristic of ester bond degradation (Figure 4b and Figure 5a) [53].

In the case of the pH-independent control, the DOX-loaded PMOXA_14_-PDMS_65_-PMOXA_14_ vesicles did not show any significant release of DOX in response to the pH lowering. The cumulative DOX loss from the vesicles was found to be only 5 ± 1% and 10 ± 1% at pH = 7.4 and pH = 3, respectively (Figure 5b). The DOX release for the non-pH-responsive vesicles at pH = 3 is similar to that for PVCL_10_-PDMS_65_-PVCL_10_ at pH = 7.4 and is mainly due to the release of DOX, which was not encapsulated into the vesicle interior but was hydrophobically bound to the outer PMOXA corona. At the pH decrease to the acidic value, DOX becomes positively charged with its solubility increased, followed by the release of the DOX traces into the solution [54].

Our data shows that the cumulative DOX release due to the degradation of the ester links between the blocks and subsequent disassembly of the PVCL-PDM-PVCL vesicles in this study was higher (96 ± 4%) than that in the case of the temperature-triggered release from the same vesicles (87 ± 10%) at 45 °C that we demonstrated earlier [36]. As mentioned above, the lower pH would increase the DOX solubility, and the drug that interacts hydrophobically with PDMS or PVCL could be released. Therefore, the dual response from PVCL_10_-PDMS_65_-PVCL_10_ polymersomes resulted in a faster and higher cumulative release of DOX. Other dually responsive block copolymers often rely on temperature to assemble nanoparticulates at temperatures higher than the LCST of a thermosensitive block [20,55,56]. In contrast, a dually responsive block copolymer, PMAA-*b*-PNIPAM, could be assembled into the nanoparticles at 45 °C and was shown to transform into nanocapsules at pH = 7 upon interaction with tannic acid (TA), utilizing hydrogen-bonding interactions between PNIPAM and TA [7].

The enhanced DOX release from PVCL-PDMS-PVCL vesicles utilizing both physiologically relevant temperature and pH is significant as it matches the environment of the cancerous tissues and can be used to shield toxic anticancer therapeutics from the normal non-pathological tissues.

### 2.4. The Effect of PVCL-PDMS-PVCL Vesicle Shell Thickness on DOX Release via the Dual Response

We also explored the effect of the PVCL-PDMS-PVCL vesicle shell thickness on the dually responsive release of DOX. We hypothesized that by changing the vesicle membrane thickness, the dually responsive release from a polymersome could be controlled. Hence, we studied the release of DOX from PVCL_5_-PDMS_30_-PVCL_5_ polymersomes with shorter PVCL and PDMS blocks synthesized according to our previous procedure [43]. We have previously shown using SANS and cryo-TEM that although PVCL_5_-PDMS_30_-PVCL_5_ has a similar hydrophilic ratio, *f* (the mass ratio of PVCL to the total copolymer weight), as PVCL_10_-PDMS_65_-PVCL_10_, with the corresponding *f* = 0.33 and *f* = 0.36, respectively, the former produces a thinner vesicle shell of 8.5 nm at 25 °C in water than the latter one, with a shell thickness of 14.6 nm under the same conditions [43].

The PVCL_5_-PDMS_30_-PVCL_5_ polymersomes were loaded with DOX (1 mg mL^−1^ DOX loading solution), dialyzed in 0.01 M phosphate buffer solution at pH = 7.4 and 25 °C until no traces of DOX in the dialysis media could be seen with UV-vis spectroscopy. The loading capacity for PVCL_5_-PDMS_30_-PVCL_5_ polymersomes from 1 mg mL^−1^ DOX solution was 34%. TEM analysis of DOX-loaded PVCL_5_-PDMS_30_-PVCL_5_ polymersomes revealed spherical vesicles that had a smaller dry size compared to that of their PVCL_10_-PDMS_65_-PVCL_10_ counterparts (Figure 6a), which is in agreement with our previous data where the average diameter of PVCL_5_-PDMS_30_-PVCL_5_ polymersomes at 25 °C was 20% smaller than that for PVCL_10_-PDMS_65_-PVCL_10_ as measured by DLS [43]. The smaller vesicle size from the shorter block copolymer (~370 nm) correlates well with a lower PVCL_5_-PDMS_30_-PVCL_5_ loading capacity of the vesicles.

The DOX release was then analyzed after the consecutive exposure of the DOX-loaded vesicles to higher temperatures (37 and 42 °C), followed by decreasing the solution acidity to pH = 3 (42 °C). Figure 6 demonstrates that there was no significant release of DOX (<10% cumulative DOX release) for at least 72 h at pH = 7.4 after the temperature was raised from 25 to 37 °C (Figure 6, circles). Raising the temperature to 42 °C resulted in the quick 27 ± 4% cumulative release of DOX within 12 h of incubation at pH = 7.4 and did not significantly increase within the 72-h incubation under these conditions (Figure 6, squares). The following incubation of this solution at a lower pH = 3 (42 °C) for an additional 48 h led to the further gradual release of DOX from the vesicles. The total DOX release reached 51 ± 3% at the 120-h time-point (Figure 6, pentagons).

Our DOX release data implies that the temperature-triggered release from PVCL_5_-PDMS_30_-PVCL_5_ polymersomes was slower than that from the vesicles made of the longer block copolymer. For instance, the shorter block copolymer vesicles released only 27% of DOX during 48-h exposure to 42 °C (pH = 7.4) (Figure 6b), while the longer PVCL_10_-PDMS_65_-PVCL_10_ copolymer vesicles released 65% of encapsulated DOX after only 10-h exposure of the vesicles to 40 °C (pH = 7.4), as we showed earlier [36]. This lower DOX release from the shorter copolymer vesicles is mainly due to less contribution of the shorter PVCL blocks to the shell shrinkage because of less dependence of PVCL_5_ on temperature [43]. This results in smaller PVCL chains pushing inward towards the PDMS layer with less contribution to the shell shrinkage and PDMS disturbance [43].

Likewise, the release of DOX at lower pH was found to be much slower from the shorter copolymer vesicles compared to the longer copolymer polymersomes. For example, the shorter copolymer vesicles released only 24% of DOX after 48 h of the low pH-treatment (Figure 6b), while 97% of DOX could be released after 24 h of the treatment (Figure 5a). These data correlate well with the slower temperature-induced release from the shorter copolymer vesicles discussed above. The less shrinkage reported for PVCL_5_-PDMS_30_-PVCL_5_ polymersomes [43] implies that increased PVCL hydrophobicity at T > 37 °C would slow down the hydrolysis of the ester links between PVCL and PDMS and subsequently hinder the DOX release, which is observed in our study (Figure 6b). These findings are important for guiding the choice of the block copolymer parameters, including the hydrophilic ratio, f, and the block copolymer length for the block copolymer design appropriate for a selective drug release with a controlled release trigger and kinetics.

### 2.5. In-Vivo Toxicity Study of PVCL_10_-PDMS_65_-PVCL_10_ Vesicle Treatment in Mice

We have demonstrated previously that empty PVCL_10_-PDMS_65_-PVCL_10_ polymersomes were not cytotoxic to cells in vitro, although the vesicles dissolved inside the acidic cellular endo/lysosomes due to degradation of the ester links between the blocks [36,49]. In this work, we evaluated the toxic potential of the PVCL-PDMS-PVCL polymersomes using a mouse model in-vivo. Sub-acute toxicity of the empty PVCL_10_-PDMS_65_-PVCL_10_ polymersomes was evaluated by intravenous injection at a single dose of 40 mg/kg of PVCL_10_-PDMS_65_-PVCL_10_ polymersome suspension administered to mice. The mice were divided into three groups that were sacrificed after 1 day (D1), 7 days (D7), and 14 days (D14) under anesthesia. The spleen, heart, kidney, and lung were collected for gravimetric and histological analyses. Mice without injection (D0) were set as control. (Table 1).

After treatment, the mice were actively observed and compared to controls. There was no mortality during the treatment and observation period, with no obvious signs of lethargic behavior. We observed that body weight at D1 was higher (*p* < 0.05) compared to the control and D7 (Table 1), which was due to the initial injection of polymersomes. A similar weight increase was also observed for the left ventricle, liver, and right kidney; however, after normalization to mice’s body weight, the differences between control, D1, and D7 were eliminated. We also noticed that body weight at D14 increased (*p* < 0.05) compared to D0 and D7, which was due to the growth of the mice, which further confirmed that mice remained healthy within 14 days post-injection (Table 1). We observed an increase (*p* < 0.05) in liver weight at D7 and D14 compared to D0; and also, the weight at D14 was higher than that at D7 (*p* < 0.05). The liver is the organ with the largest population of phagocytes within the mononuclear phagocyte system and would be expected to have the highest accumulated amount of injected nanocarriers. The observed increase in liver weight may be due to the adaptive response to the need to clear out the polymeric material. The necropsy parameters in Table 1 suggest that the PVCL-PDMS-PVCL polymersomes were well tolerated, and the increase in liver weight was not a sign of vesicle toxicity.

We also performed complete blood cell counts at D1, D7, and D14 after the injection of empty PVCL_10_-PDMS_65_-PVCL_10_ vesicles and compared them to the control untreated group (D0). Table 2 shows no significant changes in blood parameters after the mice’s treatment with polymersomes (40 mg/kg) compared to the control (D0). The hematological profile showed a decreased total leucocyte count at D14 compared to D0. We also observed statistically significant increases in counts of neutrophils, monocytes, eosinophils, platelets, and hemoglobin at D14, while only a slight decrease in lymphocytes and red blood cells at both D7 and D14. There was an increase in neutrophils for groups D7 and D14, which may be due to a mild inflammatory response. Although these changes were mild, they were statistically significant but within the range of normal values frequently observed in healthy mice.

Finally, histopathological examination was conducted on removed organs, including the heart’s left ventricle, kidney, and spleen, because they are the main organs most affected by the metabolic reaction that can be caused by a possible toxicant. The evaluation of the in-vivo effect of PVCL_10_-PDMS_65_-PVCL_10_ polymersomes on these major organs was performed at D1, D7, and D14 compared to the control (D0). Figure 7 demonstrates that the morphology of the LV, kidney, and spleen at D1, D7, and D14 showed no severe morphological changes compared to the control, without obvious necrosis or disorganized structure in the myocardium area of the heart at D14. We did not observe any signs of kidney damage (no tubular necrosis).

The hematogenic organs are very receptive to toxic byproducts and are important in revealing the pathological changes in tissues. The histological analysis of the spleen in Figure 7 showed an evidently normal tissue, with morphologically distinguishable white and red pulp. The spleen did not exhibit any enlargement, reflecting no hemolytic changes [57].

Therefore, the in-vivo toxicity study demonstrated that PVCL_10_-PDMS_65_-PVCL_10_ vesicles are safe to be further explored as a drug nanocarrier. There was no significant weight loss for 14 days post-injection of the vesicles in mice, and the major organs’ weight/body weight ratio was stable after the injection. The blood parameters remained in the healthy range during all 14 days of vesicle post-injection. The histological examination revealed no apparent pathological changes in the major organs, including the heart left ventricle, kidney, and spleen, after the subacute vesicle treatment. Overall, the in-vivo subacute toxicity study using a mouse model demonstrated good biocompatibility and non-toxicity of PVCL_10_-PDMS_65_-PVCL_10_ vesicles. These results permit further development of the vesicle system for clinical research and warrant further long-term toxicological investigation.

## 3. Materials and Methods

### 3.1. Materials

2,2′-Azobis(2-methylpropionitrile) (AIBN) (Sigma-Aldrich, Saint Louis, MO, USA) was recrystallized from methanol before use. *N*-Vinylcaprolactam (VCL) (Sigma-Aldrich) and tetrahydrofuran (THF; Fisher) were distilled prior to use. Potassium ethyl xanthate, 2-bromopropionyl bromide, diethyl ether, hexane, dichloromethane, calcium sulfate anhydrous, sodium bicarbonate, monobasic and dibasic sodium phosphate, and dialysis tubing (MWCO 3 kDa and 20 kDa) were from Fisher Scientific (Suwanee, GA, USA) and were used as received. Poly(dimetylsiloxane) bis(hydroxyethyloxypropyl) (PDMS_65_, M_n_ = 5600 Da), 2-methyl-2-oxazoline (MOXA), trifluoromethanesulfonic anhydride (Tf_2_O), potassium hydroxide, calcium chloride, diethyl ether, and methanol were purchased from Sigma-Aldrich. Potassium carbonate, sodium bicarbonate, sodium chloride, chloroform, hydrochloric acid, and glass fiber filters G4 were purchased from Fisher Scientific. Diethoxydimethylsilane, 1,3-bis(4-hydroxybutyl)tetramethyldisilane were from Silar Laboratories (Riegelwood, NC, USA). Doxorubicin hydrochloride (DOX) was purchased from LC Laboratories. Deionized (DI) water with an 18.2 MΩ cm resistivity was used for the preparation of aqueous solutions. 

### 3.2. Synthesis of Poly(N-vinylcaprolactam)_10_-block-poly(dimethylsiloxane)_65_-block-poly(N-vinylcaprolactam)_10_ (PVCL_10_-PDMS_65_-PVCL_10_) and Poly(N-vinylcaprolactam)_5_-block-poly(dimethylsiloxane)_30_-block-poly(N-vinylcaprolactam)_5_ (PVCL_5_-PDMS_30_-PVCL_5_) Triblock Copolymers

The PVCL-PDMS-PVCL triblock copolymers were synthesized by reversible addition fragment chain-transfer (RAFT) polymerization as we reported earlier [36,43,58]. For PVCL_10_-PDMS_65_-PVCL_10_ synthesis, the PDMS_65_ RAFT macroinitiator was synthesized from PDMS_65_ by a two-step reaction [36]. Briefly, PDMS_65_ with hydroxyl end groups (HO-PDMS_65_-OH) was first reacted with 2-bromopropionyl bromide in dichloromethane, followed by the reaction with potassium ethyl xanthogenate in acetonitrile, resulting in the X-PDMS_65_-X RAFT macroinitiator. The X-PDMS_65_-X macroinitiator was used for the synthesis of PVCL_10_-PDMS_65_-PVCL_10_ block copolymer via RAFT polymerization of VCL for 2 h and a feeding ratio of 1/1/80 for PDMS_65_/AIBN/monomer [43]. For PVCL_5_-PDMS_30_-PVCL_5_ triblock copolymer synthesis, PDMS_30_ with hydroxyl end groups was obtained via condensation polymerization of dimethyldimethoxysilane and 1,3-bis(4-hydroxypropyl)tetramethyldisiloxane at the same feeding ratio and reaction time of 60 min [43]. The number-average molecular weights were measured using GPC (Waters) with polystyrene linear standards (Polymer Standards Service USA Inc, Amherst, MA, USA) used for GPC calibration (*M*n = 5034 Da, Ð = 1.13 for PVCL_10_-PDMS_65_-PDMS_10_; *M*n = 3312 Da, Ð = 1.17 for PVCL_5_-PDMS_30_-PVCL_5_) [36,43].

### 3.3. Synthesis of Poly-(2-methyl-2-oxazoline)_14_-block-poly(dimethylsiloxane)_65_-block-poly-(2-methyl-2-oxazoline)_14_ (PMOXA_14_-PDMS_65_-PMOXA_14_) Triblock Copolymer

The PMOXA_14_-PDMS_65_-PMOXA_14_ block copolymer was synthesized as described previously [48]. Briefly, PDMS_65_ (10 g; 1.85 mmol) was dissolved in hexane (100 mL) in a 250 mL round-bottom flask and stirred in an ice-water bath for 30 min. Pyridine (2 g; 25 mmol) was added to the flask, and triflic acid (2.7 g; 18 mmol) was dropwise added into the flask over 15 min. After 4 h, the mixture was filtered, and the solution was washed with 1 M sodium bicarbonate solution to remove unreacted acid, and PDMS_65_-triflate was recovered by rotary evaporation. For copolymerization of PMOXA_14_-PDMS_65_-PMOXA_14_, PDMS_65_-triflate (8 g; 1.4 mmol) and 2-methyl-2-oxazoline (5 g; 59 mmol) were dissolved in dichloromethane (15 mL), and the mixture was stirred at 60 °C for 2.5 h. The reaction was quenched by adding 0.5 M KOH ethanol solution (5.5 mL) and cooling to room temperature. The number-average molecular weight of PDMS was calculated from ^1^H-NMR analysis based on the ratio between the integrals at δ = 0–0.2 ppm (SiCH_3_ protons in the PDMS block) and at δ = 0.6 ppm (methylene protons at the end group) [36,58]. An average degree of polymerization of the PMOXA block was calculated from ^1^H-NMR analysis based on the half of the ratio between the integrals at δ = 0.6 ppm (methylene proton at the end group, 4H) and at δ = 2.1~2.3 ppm (methyl protons at the carbonyl, 6H) [43]. The number-average molecular weight (*M*n = 5091 Da) and polydispersity (Ð = 1.17) of PMOXA_14_-PDMS_65_-PMOXA_14_) triblock copolymer were obtained via GPC (Waters, Milford, MA, USA) with linear polystyrene samples used as standards for GPC calibration [43].

### 3.4. Assembly of PVCL-PDMS-PVCL and PMOXA_14_-PDMS_65_-PMOXA_14_ Polymersomes

The triblock copolymer vesicles were assembled using the nanoprecipitation method [36,58]. Briefly, for nanoprecipitation, 1 mL of a copolymer ethanol solution (5 mg mL^−1^) was added dropwise to 5 mL of DI water at room temperature and stirred for 6 h, followed by the dialysis of the mixture (a Float-a-Lyzer with MWCO = 20 kDa) in DI water for 24 h to remove ethanol.

### 3.5. Loading and Release of DOX into PVCL-PDMS-PVCL and PMOXA_14_-PDMS_65_-PMOXA_14_ Vesicles

DOX was encapsulated into the vesicles using the nanoprecipitation method as reported earlier [36]. For that, 1 mL of a triblock copolymer ethanol solution (5 mg mL^−1^) was added to 5 mL of DOX aqueous solution (1 mg mL^−1^) followed by stirring for 6 h. Then, the mixture was transferred into a Float-A-Lyzer (MWCO 3 kDa) and dialyzed in DI water for 48 h to remove ethanol and non-encapsulated DOX. The loading capacity was quantified as the weight percentage of DOX in the vesicles, as analyzed by UV-visible spectroscopy (Varian Cary 50; Agilent Technologies, Santa Clara, CA, USA). For that, dialyzed DOX-loaded vesicles were lyophilized and weighed, and the encapsulated DOX was extracted by methanol. The DOX concentration was determined using absorbance spectroscopy (490 nm) and the DOX calibration curve [36]. The release of DOX from the vesicles in solutions with various pH values and temperatures was measured using the membrane dialysis method [36]. For that, a DOX-loaded vesicle solution (1.5 mL) in a Float-A-Lyzer (MWCO 3 kDa) was dialyzed in a 50 mL Falcon tube (Fisher Scientific, Suwanee, GA, USA) using 0.01 M phosphate buffer solutions at pH 7.4; 5; 4; and 3, and at 37 or 42 °C as dialysis media. Dialysis media was analyzed with UV-visible spectroscopy at various time points for DOX quantification.

### 3.6. Transmission Electron Microscopy (TEM)

TEM images of copolymer vesicles were obtained using a FEI Tecnai T12 Spirit TWIN TEM microscope (FEI Company, Hillsboro, OR, USA) operated at 80 kV. For sample preparation, a vesicle solution (7 µL) was dropped onto an argon plasma-treated Formvar/Carbon coated copper grid (200 mesh; TED Pella, Redding, CA, USA). The deposited samples were stained with 1 wt% uranyl acetate for 30 s and the solution excess was blotted off with Kimwipe paper (Fisher). 

### 3.7. Dynamic Light Scattering (DLS)

The size of PVCL_5_-PDMS_30_-PVCL_5_ and PMOXA_14_-PDMS_65_-PMOXA_14_ vesicles was measured using a Nano-ZS Zetasizer (Malvern Panalytical, Westborough, MA, USA) equipped with a He−Ne laser (663 nm) in phosphate buffer suspension at 37 °C. The average diameter of the assembled vesicles was obtained from three independent runs (15 measurements each). The particle size distribution was evaluated by standard deviation values (polydispersity index, PDI). For pH-dependent size measurements of PVCL_10_-PDMS_65_-PVCL_10_ or PMOXA_14_-PDMS_65_-PMOXA_14_ polymersomes, 1 mL of the vesicle solution (0.5 mg mL^−1^) was transferred into a Float-a-lyzer (MWCO 20 kDa) and dialyzed in 0.01 M phosphate buffer at pH = 3 in 50 mL Falcon tube (Fisher Scientific) at 37 °C. The vesicle size was then measured by DLS at 25 °C after 3, 5, and 13 h of incubation. 

### 3.8. Animal Care, Compliance, and Treatment

All animal necropsy, hematology, and histology experiment procedures were conducted according to the “Guide for the Care and Use of Laboratory Animals” (8th Edition. 2011), and AVMA Guidelines for the Euthanasia of Animals (2013 Edition) and were approved (IACUC-20356, 17/04/2017) by the Institutional Animal Care and Use Committees at the University of South Florida, Florida, USA.

### 3.9. Vesicle Toxicity Study

Risk-free C57BL/6J male mice aged 2–4 months were intraperitoneally injected with PVCL_10_-PDMS_65_-PVCL_10_ polymersomes with a dosage of 40 mg kg^−1^. The injected mice were randomly and equally divided into three groups with five mice per each group: D1 (Day 1), D7 (Day 7), and D14 (Day 14), coupled with the control group D0 (Day 0) that had not been treated with vesicle injections. Mice were sacrificed after 1 day (D1), 7 days (D7), and 14 days (D14).

### 3.10. Gravimetric Necropsy Analysis

No-injection control day (D0), D1, D7, and D14 post-injected mice were anesthetized under isoflurane briefly; then mice were injected with heparin (4 IU/g). The blood was collected from the carotid artery and centrifuged for 5 min to separate plasma. The spleen, lungs, kidney, left ventricle (LV), and right ventricle (RV) were separated and weighed individually. The LV was divided into: apex (infarcted area), mid-cavity, and base (remote area) under a microscope. The spleen and kidney were collected, weighed, and fixed in 10% zinc formalin for histology analysis.

### 3.11. Histology Analysis

The spleen, heart, and kidneys were fixed in 10% *v*/*v* formaldehyde for over 24 h and embedded in paraffin, and transversely cut into 5 mm-thick sections, followed by staining with hematoxylin and eosin (H&E staining).

### 3.12. Hematology Analysis

Blood samples were collected at euthanasia for analysis of complete blood counts (CBC). Whole blood was collected in the presence of EDTA and analyzed using a hematology analyzer (HEMAVETR 950 FS, Drew Scientific, Dallas, TX, USA).

### 3.13. Statistical Analysis

All data are expressed as mean ± standard mean error. Statistical analyses (*p* < 0.05) were performed using Graph 205 Pad Prism 7. One-way analysis of variance (ANOVA) was used. *p* < 0.05 was considered as statistically significant.

## 4. Conclusions

We synthesized poly(*N*-vinylcaprolactam)_10_-*b*-poly(dimethylsiloxane)_65_-*b*-poly(*N*-vinylcaprolactam)_10_ (PVCL_10_-PDMS_65_-PVCL_10_) and PVCL_5_-PDMS_30_-PVCL_5_ temperature-responsive triblock copolymers and assembled spherical polymer vesicles (polymersomes) in aqueous solutions using the nanoprecipitation method. An amphiphilic triblock copolymer, poly(2-methyl-2-oxazoline)_14_-*b*-poly(dimethylsiloxane)_65_-*b*-poly(2-methyl-2-oxazoline)_14_ (PMOXA_14_-PDMS_65_-PMOXA_14_) able to self-assemble into polymer vesicles were synthesized and used as a non-responsive control due to its well-studied properties. We demonstrated that temperature-responsive PVCL_10_-PDMS_65_-PVCL_10_ and PVCL_5_-PDMS_30_-PVCL_5_ vesicles could encapsulate the anticancer drug doxorubicin (DOX) with high loading capacities (40% and 34%, respectively) upon vesicle self-assembly into spherical polymersomes with average diameters of 470 and 360 nm, respectively.

Unlike the non-responsive PMOXA_14_-PDMS_65_-PMOXA_14_ vesicles, the pH-induced degradation of DOX-loaded PVCL_10_-PDMS_65_-PVCL_10_ vesicles in 0.01 M phosphate buffer solution at pH < 4 occurred within 12 h of treatment. We showed that DOX release primarily occurred at the solution acidity between 4 and 3. The 97% cumulative release of DOX from DOX-loaded PVCL_10_-PDMS_65_-PVCL_10_ vesicles was achieved in 24 h of the pH = 3 exposure of the vesicles. No significant DOX release (<10%) was observed for DOX-loaded PMOXA_14_-PDMS_65_-PMOXA_14_ vesicles after 36-h exposure of the control vesicles to pH = 3.

We also explored the effect of the PVCL-PDMS-PVCL vesicle shell thickness on the dually responsive release of DOX and demonstrated that the temperature-triggered release from PVCL_5_-PDMS_30_-PVCL_5_ polymersomes was slower than that from the vesicles made of the longer block copolymer. Likewise, the release of DOX at lower pH was found to be much slower from the shorter copolymer vesicles compared to the longer copolymer polymersomes.

We also evaluated the toxic potential of the PVCL_10_-PDMS_65_-PVCL_10_ vesicles in-vivo by intravenous injection (40 mg kg^−1^ single dose) of the vesicles into C57BL/6j male mice. The sub-acute toxicity study within 14 days included gravimetric, histological, and hematological analyses. There was no significant weight loss post-injection of the vesicles in mice, and the major organs’ weight/body weight ratio was stable for 14 days post-injection. The blood cell counts remained within the normal ranges during the post-injection 14-day period. The histology analysis did not uncover apparent pathological changes in the main organs, including the left ventricle of the heart, kidney, and spleen, after the vesicle treatment. The in-vivo toxicity examination provided evidence for the biocompatibility and non-toxicity of the PVCL-PDMS-PVCL vesicles. These results make it worthy of further insight into long-term toxicity investigation and permit further development of these vesicles for clinical research. The dually pH-/temperature-sensitive nanosized degradable polymersomes can be particularly appealing for the delivery of both hydrophilic (encapsulated inside the vesicle cavity) and hydrophobic (encapsulated within the hydrophobic membrane) anticancer therapeutics.

## Figures and Tables

**Figure 1 molecules-27-03485-f001:**
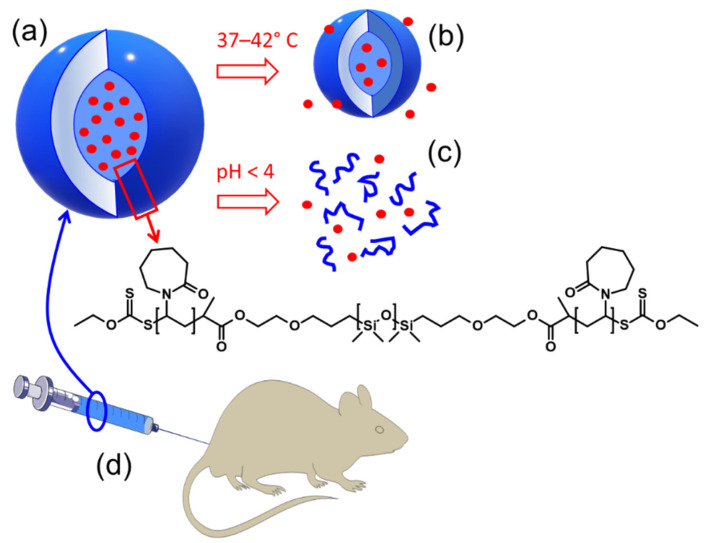
Amphiphilic PVCL_10_-PDMS_65_-PVCL_10_ triblock copolymer assembles into polymer vesicles (**a**) at room temperature and encapsulates DOX. The vesicles shrink upon temperature increase (**b**) and dissolve due to degradation of ester links between polymer blocks at acidic pH (**c**) and release the drug. In-vivo cytotoxicity of these block copolymer vesicles was evaluated using a mouse model (**d**).

**Figure 2 molecules-27-03485-f002:**
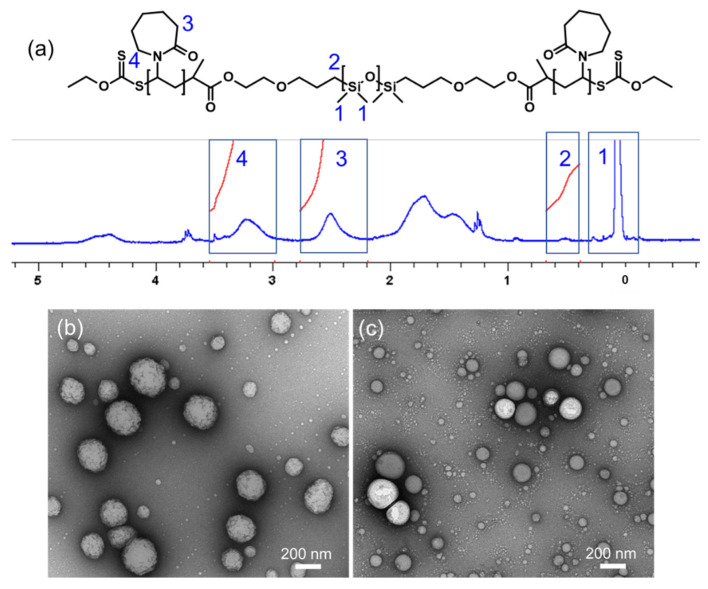
(**a**) The ^1^H-NMR spectrum of PVCL_10_-PDMS_65_-PVCL_10_. TEM images of drug (DOX)-encapsulated (1 mg mL^−1^) PVCL_10_-PDMS_65_-PVCL_10_ (**b**) and DOX-encapsulated PMOXA_14_-PDMS_65_-PMOXA_14_ (**c**) polymer vesicles.

**Figure 3 molecules-27-03485-f003:**
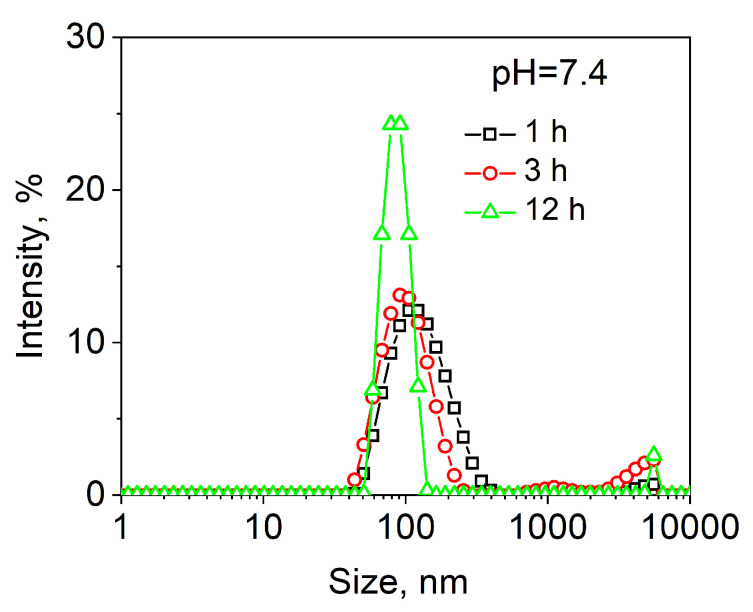
The hydrodynamic sizes of PMOXA_14_-PDMS_65_-PMOXA_14_ vesicles in the buffer solution at pH = 7.4 and 37 °C.

**Figure 4 molecules-27-03485-f004:**
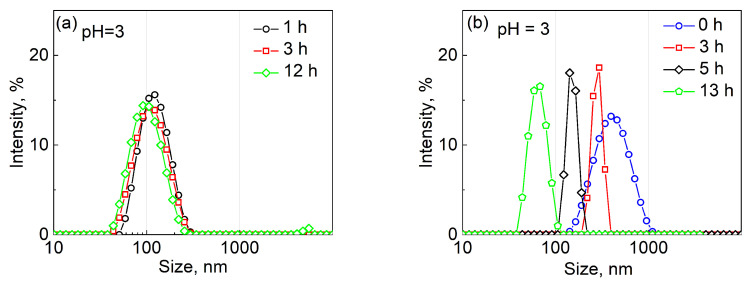
pH-dependent size of DOX-free (**a**) PMOXA_14_-PDMS_65_-PMOXA_14_ and (**b**) PVCL_10_-PDMS_65_-PVCL_10_ polymersomes in the buffer solution at pH = 3 and 37 °C after 0- (circles) 3- (squares), 5- (down-triangles), and 12-h (pentagons) incubation.

**Figure 5 molecules-27-03485-f005:**
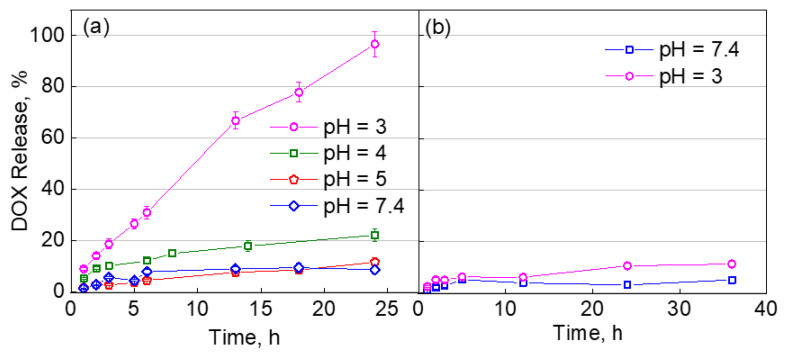
Accumulative release of DOX (%) from (**a**) DOX-loaded PVCL_10_-PDMS_65_-PVCL_10_ vesicles incubated at 37 °C and pH = 7.4 (diamonds), pH = 5 (pentagons), pH = 4 (squares), and pH = 3 (circles), and (**b**) DOX-loaded PMOXA_14_-PDMS_65_-PMOXA_14_ polymersomes at 37 °C and pH = 7.4 (squares) and pH = 3 (circles).

**Figure 6 molecules-27-03485-f006:**
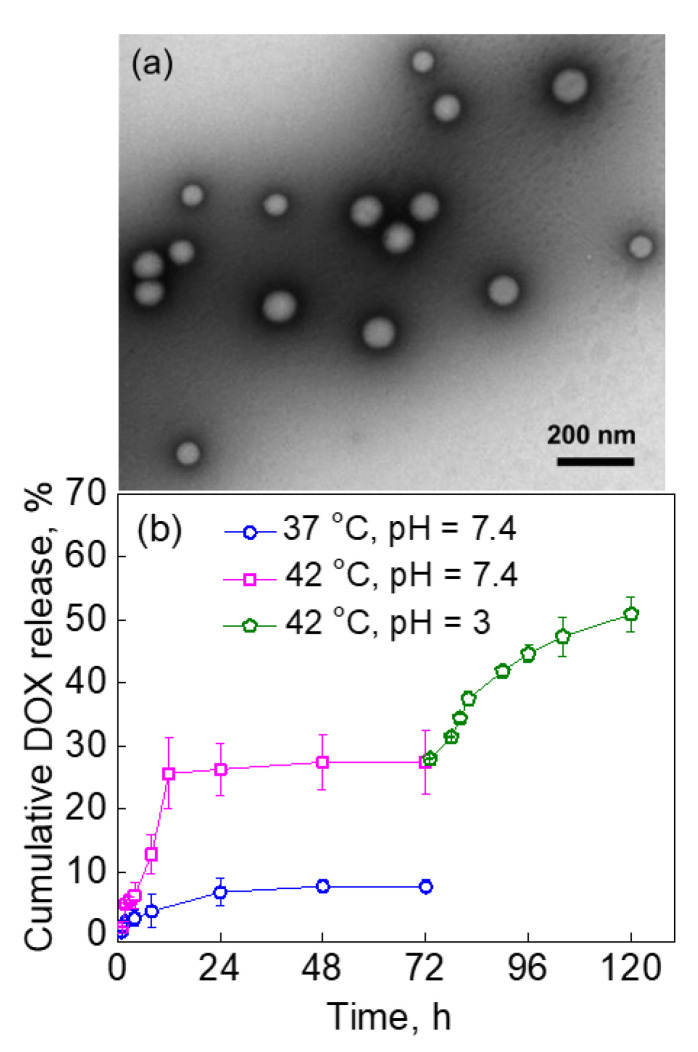
(**a**) TEM image of drug-encapsulated PVCL_5_-PDMS_30_-PVCL_5_ polymersomes. The vesicles were loaded with DOX from 1 mg mL^−1^ drug solution and had a loading capacity of 34%. (**b**) The time-dependent DOX release from DOX-loaded PVCL_5_-PDMS_30_-PVCL_5_ polymersomes at varied temperatures and pH values.

**Figure 7 molecules-27-03485-f007:**
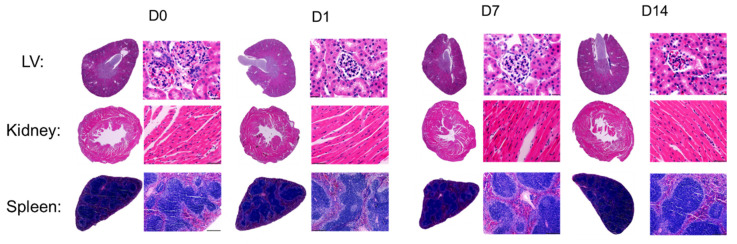
The hematoxylin and eosin (HE) staining of left ventricle (LV), kidney, and spleen post-injection with PVCL_10_-PDMS_65_-PVCL_10_ polymersomes at D0 (control), D1, D7, and D14.

**Table 1 molecules-27-03485-t001:** Necropsy Parameters in Treated and Control Mice.

Parameters	D0 (*n* = 5)	D1 (*n* = 5)	D7 (*n* = 5)	D14 (*n* = 5)
Mortality	0	0	0	0
Body weight (g)	^1^ 23.7 ± 0.4	30.3 ± 0.7 *	27.4 ± 0.5 *	32.1 ± 1.16 *
^2^ LV/^3^ BW (mg/g)	3 ± 0.14	2.9 ± 0.06	3 ± 0.04	3.1± 0.05
^4^ RV/BW (mg/g)	0.80 ± 0.04	0.7 ± 0.03 *	0.80 ± 0.02	0.8 ± 0.01
Lung weight/BW (mg/g)	5.4 ± 0.23	4.6 ± 0.13 *	4.6 ± 0.17 *	4.2 ± 0.26 *
Dry lung/BW (mg/g)	1.2 ± 0.05	1.0 ± 0.05	1.1 ± 0.05	0.9 ± 0.06
Tibia (mm)	16.2 ± 0.2	17.4 ± 0.24 *	17.1 ± 0.1 *	17.4 ± 0.02 *
LV/BW:LV/Tibia	0.2 ± 0.007	0.2 ± 0.005	0.2 ± 0.002	0.2 ± 0.002
Spleen Weight/BW (mg/g)	3.0 ± 0.30	2.6 ± 0.16	2.7 ± 0.10	2.6 ± 0.07
Right Kidney/BW (mg/g)	5.6 ± 0.11	5.2 ± 0.46	5.2 ± 0.12	5.6 ± 0.14
Left Kidney/BW (mg/g)	5.5 ± 0.24	5.3 ± 0.43	5.3 ± 0.08	5.3 ± 0.21
Liver/BW (mg/g)	41.5 ± 2.36	40.0 ± 1.85	44.5 ± 1.69	42.12 ± 0.77

^1^ Values are mean ± standard mean error; *n* indicates sample size. ^2^ LV, left ventricle; ^3^ BW, bodyweight; ^4^ RV, right ventricle; mice were sacrificed on the same day (D0), after 1 (D1), 7 (D7) and 14 days (D14) after treatment; * *p* < 0.05 vs. D0 control. All organ weights are normalized to body weight.

**Table 2 molecules-27-03485-t002:** Hematological Profile of Treated and Control Mice.

Hematological Profile	D0 (*n* = 5) ^1^	D7 (*n* = 5) ^1^	D14 (*n* = 14) ^1^	Normal
Total leucocytes (K/µL)	4.70 ± 0.46	5.4 ± 0.87	3.74 ± 0.44 *	1.8–10.7
Neutrophils (%)	9.55 ± 1.02	21.55 ± 2.45 *	20.58 ± 2.08 *	6.6–38.9
Lymphocytes (%)	86.38 ± 1.18	71.81 ± 2.26 *	72.43 ± 1.94 *	55.8–91.6
Monocytes (%)	3.77 ± 0.785	5.87 ± 0.15 *	5.56 ± 0.82 *	0.0–7.5
Eosinophils (%)	0.27 ± 0.13	0.66 ± 0.23 *	1.12 ± 0.59 *	0.0–3.9
Platelets (K/µL)	741.25 ± 31.66	665.8 ± 149.68	905.8 ± 23.31 *	592–2972
Red Blood Cells (M/µL)	9.042 ± 0.24	8.91 ± 0.09	8.58 ± 0.09 *	6.36–9.42
Hemoglobin (g/dL)	10.88 ± 0.40	11.84 ± 0.12 *	10.28 ± 0.13 *	11.0–15.1
Hematocrit (%)	40.18 ± 1.09	39.58 ± 0.37	37.02 ± 0.55 *	35.1–45.4

^1^ Values are mean ± standard mean error; *n* indicates sample size.; mice sacrificed on Day 7 (D7) and Day 14 (D14) were compared to Day 0 (D0) as naïve controls; * *p* < 0.05 vs. D0 control.

## Data Availability

The data presented in this study are available on request from the corresponding author.

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
