# Peer review of "Dually Responsive Poly(N-vinylcaprolactam)-b-poly(dimethylsiloxane)-b-poly(N-vinylcaprolactam) Polymersomes for Controlled Delivery"

_molecules, 2022, doi:10.3390/molecules27113485_

Round 1

Reviewer 1 Report

The text is correctly edited, the abstract, the introduction, research results, methods and summary are properly written. The results of the research, compared with the results of other authors, are a valuable base for writing scientific papers.

My suggestions for improvement:

line 511: "naïve" - I don't understand that word

line 603: "Sigma-Aldrich.," - remove the comma

lines 644, 651: [Error! Bookmark not defined.] - please correct

The work is extensive so slight editorial errors may occur, however they do not significantly affect the quality of the manuscript.

Author Response

We express our deep gratitude to the reviewer for their thorough and attentive reading of our manuscript. We considered all provided suggestions and comments and made appropriate corrections with explanations, which improved the quality of our revised paper. Our responses follow below.

The text is correctly edited, the abstract, the introduction, research results, methods and summary are properly written. The results of the research, compared with the results of other authors, are a valuable base for writing scientific papers.

My suggestions for improvement:

line 511: "naïve" - I don't understand that word

In cell or animal studies, this word is used as a technical term to reflect the maturity of the cell or an animal that, however, has not yet encountered any treatment. This term has been removed from the revised manuscript to avoid any possible confusion.

line 603: "Sigma-Aldrich.," - remove the comma

The non-essential comma has been removed in the revised paper.

lines 644, 651: [Error! Bookmark not defined.] - please correct The work is extensive so slight editorial errors may occur, however they do not significantly affect the quality of the manuscript.

We thank the reviewer for the attentive reading of our paper. We have corrected the missing references in the revised manuscript.

Reviewer 2 Report

This is an interesting study about dually responsive polymersomes for controlled drug delivery. I recommend it for publication in Molecules after the following issues are addressed.

  1. Line 56, slightly lowered pH in acidic tumor, inflamed tissues, and in the endosomal and lysosomal intracellular compartments should not be as low as to pH 3. The authors should add more references about this point.
  2. In the introduction, 'Among those, polymer vesicles assembled from block copolymers, i.e., polymersomes, have been recognized as effective nanocarriers...' Several more recent studies (Biomacromolecules 2020, 21, 8, 3353–3363; Langmuir 2019, 35, 5, 1273–1283; Acta Biomaterialia 80 (2018): 327-340) related to polymersome for drug delivery should be included. 
  3. It is not clear PDMS was chosen as the hydrophobic block.
  4. Figure 3, why PMOXA14-PDMS65-PMOXA14 vesicles were unstable in pH 7.4 buffer?
  5. Figure 2 and 6, why drug-loaded polymersomes seem to be more purfied than drug-free polymersomes? 
  6. The PDI of the polymers used in this study should be added.

Author Response

We express our deep gratitude to the reviewer for their thorough and attentive reading of our manuscript. We considered all provided suggestions and comments and made appropriate corrections with explanations, which improved the quality of our revised paper. Our responses follow below.

  1. Line 56, slightly lowered pH in acidic tumor, inflamed tissues, and in the endosomal and lysosomal intracellular compartments should not be as low as to pH 3. The authors should add more references about this point.

We thank the reviewer for noticing this issue. We have corrected this statement and added two more references. The revised discussion now reads as follows:

The use of dually-responsive polymersomes that are assembled from temperature- and pH-sensitive block copolymers are particularly interesting due to the slightly lowered acidic pH (pH = 6.8-5) [22] and elevated temperature (T= 38-41 42 °C) [23] conditions encountered in the tumor, inflamed tissues, and in the endosomal and lysosomal intra-cellular compartments [24].

  1. In the introduction, 'Among those, polymer vesicles assembled from block copolymers, i.e., polymersomes, have been recognized as effective nanocarriers...' Several more recent studies (Biomacromolecules 2020, 21, 8, 3353–3363; Langmuir 2019, 35, 5, 1273–1283; Acta Biomaterialia 80 (2018): 327-340) related to polymersome for drug delivery should be included.

We thank the reviewer for bringing up recent relevant references. We have included two of the suggested references (Biomacromolecules, Acta Biomaterialia) instead of outdated references 10,13 in the revised manuscript.

  1. It is not clear PDMS was chosen as the hydrophobic block.

We thank the reviewer for this comment. We have revised the Introduction to include the statement about PDMS and PVCL general biocompatibility with the corresponding references, which now reads as follows: “PDMS and PVCL are both biocompatible and generally not cytotoxic [37,38,39].”

  1. Figure 3, why PMOXA14-PDMS65-PMOXA14 vesicles were unstable in pH 7.4 buffer?

Figure 3 shows time-dependent DLS measurements of the PMOXA14-PDMS65-PMOXA14 vesicles at pH = 7.4. The data show that these vesicles were stable in solution at pH = 7.4 for at least 3 hours. However, within the 12-hour period, we observed a slight size decrease to an average diameter of 91 ± 60 nm, which was accompanied by increased scattering intensity at a micrometer size range (> 1 µm) (Figure 3, triangles). This result can be explained by a slow aggregation of the vesicles with time at this pH and settling down as reported recently [45], which agrees with the data on the average size and size distribution decrease. This explanation is included in the Discussion as lines 258-271.    

  1. Figure 2 and 6, why drug-loaded polymersomes seem to be more purified than drug-free polymersomes?

We have checked the TEM images in Figure 2 and 6. In both images, we show DOX-loaded PVCL-PDMS-PVCL vesicles assembled via the nanoprecipitation method. However, in Figure 2, the vesicles are assembled from a longer block copolymer of PVCL10-PDMS65-PVCL10, while in Figure 6, the vesicles are obtained from a shorter block copolymer of PVCL5-PDMS30-PVCL5. In both cases, the polymersomes were purified using the same dialysis protocol. We did not encounter any signs of impurities in neither case.

  1. The PDI of the polymers used in this study should be added.

We thank the reviewer for pointing out this omission. We have added the information on the polymers’ number-average molecular weights and polydispersity as follows:

Line 627-630: The number-average molecular weights were measured using GPC (Waters) with poly-styrene linear standards (Polymer Standards Service USA Inc) used for GPC calibration (Mn = 5034 Da, Ð = 1.13 for PVCL10-PDMS65-PDMS10; Mn = 3312 Da, Ð = 1.17 for PVCL5-PDMS30-PVCL5) [36,43].

And Line 649-651: The number-average molecular weight (Mn = 5091 Da), polydispersity (Ð = 1.17) of PMOXA14-PDMS65-PMOXA14) triblock copolymer was obtained via GPC (Waters) with linear polystyrene samples used as standards for GPC calibration [43].

Reviewer 3 Report

The paper submitted by Kozlovskaya et al. deals with the preparation of polymersomes starting from different block copolymers such as: PNVCL-PDMS-PNVCL and PMOXA-PDMS-PMOXA. These polymersomes were loaded with DOX and then characterized as a function of pH and temperature by different methods. The in-vivo tests demonstrated that these systems can be used as safe drug delivery systems.

The manuscript is clear, well written and the conclusions are supported by the results. However, some corrections are needed in order to increase the overall quality of the paper:

  1. the introduction section should be completed with other papers concerning the use of PNVCL-based block copolymers. A suggestion can be: https://doi.org/10.1016/j.eurpolymj.2019.07.015
  2. in order to demonstrate the presence of different interactions between the drug and the polymeric matrix, the authors must carry out some FTIR analysis.
  3. TEM photos of the drug-loaded polymersomes must also be added
  4. the authors state that the higher drug release rate from the polymersomes having longer PNVCL chains is related to the shrinkage of the PNVCL chains but they also need to consider the hypothesis that the amount of DOX physically adsorbed on these longer PNVCL chains is higher than that adsorbed on PNVCL having a DP of 5.
  5. please revise the caption of fig. 2
  6. L218: delete the first "and"
  7. L715: "An polymer...was synthesized..."

Author Response

We express our deep gratitude to the reviewer for their thorough and attentive reading of our manuscript. We considered all provided suggestions and comments and made appropriate corrections with explanations, which improved the quality of our revised paper. Our responses follow below.

The paper submitted by Kozlovskaya et al. deals with the preparation of polymersomes starting from different block copolymers such as: PNVCL-PDMS-PNVCL and PMOXA-PDMS-PMOXA. These polymersomes were loaded with DOX and then characterized as a function of pH and temperature by different methods. The in-vivo tests demonstrated that these systems can be used as safe drug delivery systems.

The manuscript is clear, well written and the conclusions are supported by the results. However, some corrections are needed in order to increase the overall quality of the paper:

  1.   the introduction section should be completed with other papers concerning the use of PNVCL-based block copolymers. A suggestion can be: https://doi.org/10.1016/j.eurpolymj.2019.07.015

We thank the reviewer for the suggested reference. We have revised this part of Introduction to refer to papers on PVCL-based materials. It now reads as follows (lines 89-93): “PDMS and PVCL are both biocompatible and generally not cytotoxic [37,38,39]. PVCL is a temperature-sensitive polymer with a phase transition in the range from 36 to 50 °C. The development of the PVCL-based self-assemblies for controlled drug delivery has been gaining growing attention in recent years [40,41,42]. We showed that these PVCL-PDMS-PVCL polymersomes reversibly decreased in size without changing their shape upon the solution temperature increase in the range from 37 to 42 °C due to the lower critical solution temperature (LCST) transition of the PVCL blocks [36].”

  1. in order to demonstrate the presence of different interactions between the drug and the polymeric matrix, the authors must carry out some FTIR analysis.

We thank the reviewer for this suggestion. The study on the interactions between the drug and the polymer chains is beyond the scope of this work. This work is focused on dual pH- and T-responses from the PVCL-PDMS-PVCL polymersomes and their toxicity potential in-vivo. However, it might be an interesting aspect of this system to explore in our future work.

  1. TEM photos of the drug-loaded polymersomes must also be added.

We thank the reviewer for this suggestion. The TEM images of DOX-loaded PVCL-PDMS-PVCL vesicles are presented in the manuscript in Figure 2 (DOX-loaded PVCL10-PDMS65-PVCL10 vesicles) and in Figure 6 (DOX-loaded PVCL5-PDMS30-PVCL5 vesicles). We have revised both figure captions to clarify that in both cases DOX-loaded vesicles are shown.

  1. the authors state that the higher drug release rate from the polymersomes having longer PNVCL chains is related to the shrinkage of the PNVCL chains but they also need to consider the hypothesis that the amount of DOX physically adsorbed on these longer PNVCL chains is higher than that adsorbed on PNVCL having a DP of 5.

We thank the reviewer for this interesting observation. We report the cumulative DOX release in percentage (%) and not in the absolute values. DOX is primarily encapsulated into the vesicle’s interior cavity, and it can be less adsorbed on the PVCL or PDMS chains. Due to this, the release rate will be mainly dictated by the permeability of DOX through the vesicle membrane at 37-42 °C. The diffusion of DOX through PDMS membrane is observed when PVCL chains collapse and insert themselves within PDMS membrane as we showed in our previous work. Shorter PVCL5 will not shrink much and thus will not disturb the PDMS membrane as much as PVCL10 would do, hence the less amount of the released DOX from shorter PVCL vesicles.

  1. please revise the caption of fig. 2

The Figure 2 caption has been revised as follows: “Figure 2. (a) The 1H NMR spectrum of PVCL10-PDMS65-PVCL10. TEM images of drug (DOX)-encapsulated (1 mg mL-1) PVCL10-PDMS65-PVCL10 (b) and DOX-encapsulated PMOXA14-PDMS65-PMOXA14 (b) polymer vesicles”

  1. L218: delete the first "and"

This sentence has been revised as follows: “After 6-hour assembly and DOX encapsulation, the vesicles were purified by dialysis in 0.01 M buffer solution at pH = 7.4 to remove ethanol and free, nonencapsulated DOX.”

  1. L715: "An polymer...was synthesized..."

This issue has been resolved.